# The Role of Gut Microbiota and Leaky Gut in the Pathogenesis of Food Allergy

**DOI:** 10.3390/nu16010092

**Published:** 2023-12-27

**Authors:** Remo Poto, William Fusco, Emanuele Rinninella, Marco Cintoni, Francesco Kaitsas, Pauline Raoul, Cristiano Caruso, Maria Cristina Mele, Gilda Varricchi, Antonio Gasbarrini, Giovanni Cammarota, Gianluca Ianiro

**Affiliations:** 1Department of Translational Medical Sciences, University of Naples Federico II, 80131 Naples, Italy; remo.poto@gmail.com (R.P.); gildanet@gmail.com (G.V.); 2Center for Basic and Clinical Immunology Research (CISI), University of Naples Federico II, 80131 Naples, Italy; 3World Allergy Organization (WAO), Center of Excellence, 80131 Naples, Italy; 4Department of Translational Medicine and Surgery, Università Cattolica del Sacro Cuore, 00168 Rome, Italyemanuele.rinninella@unicatt.it (E.R.); marco.cintoni@unicatt.it (M.C.); francesco.kaitsas01@icatt.it (F.K.); pauline.raoul1@gmail.com (P.R.); cristiano.caruso@policlinicogemelli.it (C.C.); mariacristina.mele@policlinicogemelli.it (M.C.M.); antonio.gasbarrini@unicatt.it (A.G.); giovanni.cammarota@unicatt.it (G.C.); 5Department of Medical and Surgical Sciences, UOC Gastroenterologia, Fondazione Policlinico Universitario Agostino Gemelli IRCCS, 00168 Rome, Italy; 6Department of Medical and Surgical Sciences, UOSD DH Internal Medicine and Digestive Diseases, Fondazione Policlinico Universitario Agostino Gemelli IRCCS, 00168 Rome, Italy; 7Department of Medical and Surgical Sciences, Clinical Nutrition Unit, Fondazione Policlinico Universitario Agostino Gemelli IRCCS, 00168 Rome, Italy

**Keywords:** food allergy, intestinal barrier, leaky gut, gut microbiota, prebiotics, probiotics, synbiotics, fecal microbiota transplantation

## Abstract

Food allergy (FA) is a growing public health concern, with an increasing prevalence in Western countries. Increasing evidence suggests that the balance of human gut microbiota and the integrity of our intestinal barrier may play roles in the development of FA. Environmental factors, including industrialization and consumption of highly processed food, can contribute to altering the gut microbiota and the intestinal barrier, increasing the susceptibility to allergic sensitization. Compositional and functional alterations to the gut microbiome have also been associated with FA. In addition, increased permeability of the gut barrier allows the translocation of allergenic molecules, triggering Th2 immune responses. Preclinical and clinical studies have highlighted the potential of probiotics, prebiotics, and postbiotics in the prevention and treatment of FA through enhancing gut barrier function and promoting the restoration of healthy gut microbiota. Finally, fecal microbiota transplantation (FMT) is now being explored as a promising therapeutic strategy to prevent FA in both experimental and clinical studies. In this review article, we aim to explore the complex interplay between intestinal permeability and gut microbiota in the development of FA, as well as depict potential therapeutic strategies.

## 1. Introduction to Leaky Gut and Gut Microbiota in Food Allergies

### 1.1. Gut Microbiota

The term “gut microbiota” refers to the complex community of microbes that inhabit the gut, which includes bacteria, fungi, viruses, and parasites [1]. Four major phyla, Firmicutes, Bacteroidetes, Actinobacteria and Proteobacteria, account for more than 90% of the whole bacteriome [2]. The composition of the gut microbiota varies from site to site, and it is extremely dynamic, depending on a huge number of factors, including age, diet, habits, drug use, and others [3,4,5,6]. Gut microbiota play a key role in human health and disease. As alterations to their composition and structure, generically defined as dysbiosis, may lead to diseases, a healthy microbiota helps in promoting and maintaining gut health, through a trophic effect on enterocytes, by maintaining intestinal barrier integrity, avoiding the adhesion of pathogenic bacteria, and producing vitamins [7,8,9]. Gut microbiota are also connected to other regions of the body, having complex metabolic systemic effects [10,11,12] and even interacting with the nervous system thanks to the so-called gut–brain axis [13,14].

Increasing evidence supports the role of gut microbiota in the pathogenesis of food allergies (FA) [15]. First, gut microbiota play a role in maintaining intestinal barrier integrity, as well as in modulating the gut inflammatory microenvironment. Moreover, modulation of the gut microbiota, from pre- and probiotics to fecal microbiota transplantation (FMT), is opening new and previously unknown possibilities for the treatment of conditions whose prevalence and social impact grows daily [16,17,18].

### 1.2. Intestinal Barrier

The primary function of the intestinal barrier is to facilitate the absorption of nutrients and fluids, while blocking the entry of harmful substances, such as toxins and pathogens, into the underlying tissue through the intestinal epithelium [19]. Moreover, it plays a vital role in ensuring the overall fitness and health of the body, protecting us from potentially harmful agents coming from the gut, and being the main controller of the substances that can be absorbed and the ones that should be blocked [20].

The intestinal barrier consists of multiple components. The outer layer includes various extracellular components, whose main actor is the mucus layer. It is composed of highly glycosylated mucin proteins, i.e., MUC2 [21]. The mucus layer contains an outer layer called the “stirred mucus layer” that acts as a first line of defense, due to the presence of secretory immunoglobulin A (sIgA) and antimicrobial products [22]. There is also an inner layer called the “non-stirred mucus layer” or “glycocalyx”, which is mainly devoted to nutrient absorption and epithelial protection, and cellular renewal and differentiation [23]. The intestinal barrier is also made of extracellular elements, including gut microbiota and digestive enzymes, such as proteases, lipases, amylases, and nucleases [24]. The inner layer of the intestinal barrier consists of epithelial cells that create a cohesive and polarized single layer that strictly controls the passage of molecules, through the aid of junctional complexes better described below in the text, carrying out both digestive and immune functions [25]. The intestinal barrier is mainly made up of enterocytes, but goblet cells, Paneth cells, enterochromaffin cells, and M cells can also be found. The pluripotent stem cells, which are important for the renewal of these elements, are located within the Lieberkühn crypts [26]. Underneath these two layers, there is the lamina propria, which contains innate and adaptive immune cells that build the gut-associated lymphoid tissue (GALT) [27], the connective tissue, and the enteric nervous system, composed by the myenteric (Auerbach’s) plexus and the submucosal (Meissner’s) plexus, which control motor and secretory functions [28]. Three main pathways allow molecules to cross the epithelial layer: the paracellular pathway, which allows the passage of substances through the intercellular space between cells; the carrier-mediated pathway, where substances are transported through the aid of carrier proteins; and the trans-cellular pathway, where substances pass through the cell [29].

Specialized areas of contact between the plasma membranes of neighboring cells are known as intercellular junctions. There are three main types of intercellular junctions: tight junctions, which are occluding junctions that enable the establishment and maintenance of concentration gradients across an epithelium [30]; anchoring junctions, which facilitate the adhesion of cells to either other cells or the substrate, enabling the distribution of local tensile forces across a tissue [31]; and GAP junctions, which facilitate communication between adjacent cells, enabling the exchange of ions or small molecules between closely connected cells [32].

### 1.3. Food Allergies

FAs are pathological immune reactions triggered by normally innocuous food protein antigens [33]. Their prevalence rises every year [34], and it is estimated to be 5% for adults and 8% for children [35]; nevertheless, the significative cost for healthcare systems [36], the impact on the personal and social life of the patients, and findings about new possible therapies make FA a relevant and current topic.

FAs must be distinct from food intolerances, as in those conditions the immune system is not involved, while coeliac disease is not typically classified as a food-allergic disease [33]. FAs may be classified according to their immunopathogenic pathway into three categories. The IgE-mediated FA group sees food antigens binding to preformed IgE, triggering a type-1 hypersensitivity reaction with degranulation of mast cells and basophils, which results in rapid appearance of symptoms, usually within minutes [37]. The not-IgE mediated group is characterized by the action of allergen-specific T helper 2 cells, with more delayed symptoms [38]. Eventually, mixed FA form the third group, in which IgE-dependent and IgE-independent pathways coexist; patients suffering from this kind of allergy are usually affected by eosinophilic gastrointestinal disorders, such as eosinophilic esophagitis [33].

Food allergic reactions are only possible if a previous sensitization, defined as the presence of food-specific IgE, has already happened [39]. In normal conditions, the antigens that reach the intestinal barrier are unable to pass through it directly. However, they cross it thanks to cell-mediated mechanisms, like the action of goblet cells or CX3CR1 macrophages, which transfer the antigens to dendritic cells and those to draining lymph nodes [40,41], promoting the differentiation of T-naïve cells into food-antigen specific T regulatory cells [42]. When the epithelial barrier is damaged, antigens, as well as pathogens, may cross it freely, and this leads to the release of pro-inflammatory molecules like IL-25, IL-33, and thymic stromal lymphopoietin (TSLP), which promote T-naïve cell differentiation into T H2 cells, IgE class-switching, and tissue accumulation of mast cells and eosinophils [43,44,45]. Once sensitization has occurred, allergic reactions to that specific food antigen may occur, with symptoms ranging from cutaneous rashes to persistent diarrhea and potentially deadly anaphylaxis [37].

Gut microbiota are one of the main actors in the complex mechanism of sensitization [46,47]. Low microbial diversity and an elevated Enterobacteriaceae/Bacteroidaceae ratio are associated with subsequent food sensitization in children [48], and children born through caesarean (who do not acquire the mother’s vaginal microbiota) are known to be at higher risk of allergic diseases [49]. Moreover, bacterial metabolites like short-chain fatty acids (SCFAs) are also involved. SCFAs are end products of bacterial fermentation, mostly produced by Firmicutes, the most relevant of which are butyrate and propionate [50]. These molecules exert anti-inflammatory effects that improve the epithelial barrier integrity, reducing the possible sensitization to food allergens [51,52].

At present, avoidance of the food allergen and treatment of the allergic reactions remain the milestone of FA management [53]. Nevertheless, immunotherapy has opened the possibility of a real “definitive” treatment [54,55]; and prevention strategies such as the early assumption of specific foods, like peanuts, are increasingly coming into practice [56,57]. Among these findings, the role of microbiota as a key-player in the gut barrier has emerged as a new area of research and possible treatment for these conditions. In the next chapters, we will more deeply explore its involvement in FA pathophysiology, and how its modulation may relieve these conditions.

### 1.4. Leaky Gut

The term “leaky gut” defines a status of weakening or disruption of the intestinal barrier, in which substances that are normally confined in the intestinal cavity pass through the intestinal wall and enter the bloodstream [58]. The disruption of the intestinal barrier and the increase in the blood concentration of toxins have both local and systemic consequences [59,60]. First, an increased immune activation in the lamina propria through the pathway of Toll-like Receptor (TLR4) has been reported, whose ligand is mainly LPS [61]. By binding LPS to its receptor TLR4, the NF-κB pathway is activated, raising the concentration of pro-inflammatory cytokines [62]. This pro-inflammatory environment increases the amount of immune cells in various organs such as the liver, adipose tissue, and muscles, enhancing insulin resistance [63] and fostering the process of atherosclerosis [64].

Many factors can lead to an increase in intestinal permeability, including diet, inflammatory conditions like inflammatory bowel diseases (IBD), cirrhosis and its complications, burn injuries [65], and alcohol [66] through different pathophysiological pathways [67]. Specific alterations in daily diet can alter gut permeability. Reduced presence of the vitamin D receptor causes downregulation of claudin-2, a protein involved in the formation of tight junctions, increasing gut leakiness [68].

Moreover, deficiency in dietary fibers lets the microbiota feed on mucus glycoproteins of the host, disrupting the muco-epithelial barrier and increasing intestinal permeability [69]. In addition, a diet high in saturated fats decreases the abundance of *Lactobacilli,* while increasing the number of *Oscillospiraceae*, which is negatively correlated with the mRNA expression of Zonulin-1 (ZO-1), a barrier-forming TJ protein, leading to the condition of leaky gut [70].

Because of the disruption of the gut barrier, IBDs are characterized by a considerably leaky gut [71,72]. Patients with IBD experience both a loss in the number of tight junctions and a qualitative difference in their structure [73], as well as presenting altered types of intestinal cells. First, a reduced number of goblet cells can be observed in the epithelium of patients with IBD, and these cells also produce a minor amount of mucus [74]. Moreover, Paneth cells can be found in colonic mucosa and not only in the Lieberkühn crypts, resulting in an increase in the secretion of protective proteins in the large intestine [75]. This condition of intestinal inflammation and the disruption of the gut barrier result in an altered passage of substances in blood circulation, leading to systemic proinflammatory conditions [76].

Intestinal permeability is also altered in liver disorders, mainly in cirrhosis [77]. Due to the close relationship between gut and liver, because of the portal circulation, a large amount of pathogen-associated molecular patterns (PAMPs) reach the liver and increase its inflammation, speeding up liver diseases [78]. Patients with cirrhosis present different microbial profiling, with an increase in the abundance of bacteria in the phyla Proteobacteria and Fusobacteria and a reduction in bacteria in the phylum Bacteroidetes [79].

These alterations to gut microbiome, in addition to a reduced intestinal permeability, boost liver inflammation, which worsens the illness and its complications, such as spontaneous bacterial peritonitis (SBP) [80] and hepatic encephalopathy [81].

Thus, leaky gut plays a key role in the pathogenesis of many diseases, many of them being autoimmune diseases or with a disimmune pathogenetical component, like IBD, coeliac disease, autoimmune hepatitis, and multiple sclerosis [71,82,83,84]. The release of proinflammatory molecules and epithelial damage are factors shared by essentially all of these diseases [85]. In particular, the inflammatory pathway of FA, differently from other disorders, is characterized by the overactivation of the Th2 pathway and relative cytokines, like IL-4 and IL-13, as further explained [86].

### 1.5. Leaky Gut, Gut Microbiome, and Implications in Food Allergies

Leaky gut has garnered significant interest due to its potential role in the onset of FA. The “epithelial barrier hypothesis” assumes that intestinal barrier dysfunction, which increases susceptibility to environmental factors, can contribute to the sensitization and development of allergic diseases [87]. This hypothesis suggests that alterations in the intestinal integrity, often associated with leaky gut, might play a role in the observed rise in FA prevalence [88]. Environmental and lifestyle factors, such as industrialization and the consumption of ultra-processed foods (UPFs), are believed to contribute to disrupting intestinal barrier integrity [87]. These external factors, which constitute the external exposome, can influence the gut microbiome and epithelial barriers, playing a significant role in the development of allergic diseases [89]. Food emulsifiers in UPFs (i.e., polysorbate 20 and 80) have a detrimental effect on intestinal epithelial integrity [90]. Reduced levels of antioxidants and vitamins in UPFs are thought to increase susceptibility to allergic diseases [89]. Furthermore, it has been hypothesized that increased exposure to advanced glycation end products (AGEs) may be linked to the increased prevalence of FA [91].

A compromised epithelial barrier can trigger immune responses that result in the release of inflammatory mediators, known as alarmins [92]. Alarmins are epithelial-derived cytokines, such as thymic stromal lymphopoietin (TSLP), interleukin 33 (IL-33), and IL-25, released in response to the cellular damage caused by cellular stress or infection [93,94,95]. While these cytokines have crucial roles in maintaining gut epithelial homeostasis, they can also promote a pro-allergic microenvironment by activating T helper 2 (Figure 1 [96,97,98,99,100,101,102,103]) and type 2 innate lymphoid cells [99,104,105,106,107,108,109,110].

Several studies in mice and humans have linked increased intestinal barrier permeability to FA [111,112,113,114,115,116]. This increased permeability allows the translocation of allergenic molecules, such as dietary proteins, toxins, and microbial products, across the intestinal epithelium, where they can interact with immune cells in the gut-associated lymphoid tissue [117]. As a result, an aberrant immune response is triggered, leading to the production of allergen-specific immunoglobulin E (IgE) antibodies and subsequent activation of mast cells and basophils. Upon re-exposure to the allergen, IgE antibodies bind to the surface of mast cells and basophils, releasing proteases and inflammatory mediators, such as histamine [95,118]. This immunological response further impacts intestinal permeability, and the increase in allergen passage perpetuates the inflammatory response through immediate hypersensitivity reactions. Indeed, intestinal permeability may increase when intestinal mucosal barrier function is impaired during allergy [119,120,121].

A breach in the epithelial barrier may result from Th2 immune responses or exaggerated reactions caused by mast cell activation. Type-2 cytokines, such as IL-4 and IL-13, have been shown to directly affect the permeability of intestinal epithelial cells [86,122,123]. These cytokines reduce transepithelial resistance and increase the movement of macromolecules across the epithelium [86,122]. They also influence the expression of tight junction proteins, leading to enhanced permeability [124]. In sensitized mice, ovalbumin (OVA) downregulates the expression of zonulin (ZO)-1, which plays an important role in the regulation of tight junction permeability [125,126].

When the intestinal barrier becomes compromised and “leaky”, it allows increased permeability of the gut epithelial layer [127]. This means that allergenic proteins in ingested food have a higher chance of crossing this compromised barrier and entering the bloodstream. Several studies have demonstrated that individuals with FA have a distinctive transcellular facilitated transport route for the transportation of allergens [127]. This protein uptake has been demonstrated to be specific and exclusively transcellular, occurring within minutes of the allergen exposure and facilitated by the IgE/CD23 complex [128,129,130]. In contrast, healthy individuals primarily rely on transcytosis through enterocytes as the main pathway for protein uptake [131,132]. Moreover, it has been observed that, in FA individuals, the release of mediators from mast cells leads to an augmented allergen transport via the paracellular route [127].

Psychological stress and mast cell activity may also play a role in gut barrier defects and intestinal sensitization [133]. However, it is still unclear whether intestinal barrier dysfunction is the main initiating factor for FA sensitization [134]. A recent study indicates that a gastrointestinal bacterial infection can potentially disrupt oral tolerance to dietary antigens in mice [135]. This disruption can trigger an adaptive immune response against food antigens, resulting in increased intestinal permeability and abnormal pain signaling upon subsequent exposure to the same antigens [135]. Further research using experimental models and antigens with protease activity is needed to determine the order of events between permeability changes and allergic sensitization.

Emerging evidence suggests that compositional and functional changes to the gut microbiome, also known as dysbiosis, contribute to the development and progression of FA, as happens for other diseases [82,136,137,138]. Early studies using germ-free mice have provided initial insights into the significance of the gut microbiota in modulating FA. These studies demonstrated that germ-free mice were unable to achieve oral tolerance to food allergens, and successful induction of oral tolerance through intestinal microbiota reconstitution was only feasible in neonatal mice [139]. Recent studies have further shown that transferring gut microbiota from patients with FA to germ-free mice can transmit susceptibility to FA [140,141]. On the other hand, germ-free mice colonized with bacteria from healthy, but not FA, infants were protected against anaphylactic responses to a cow milk allergen [141]. Multiple microbial orders, including *Clostridiales* and *Lactobacillales*, have been implicated in FA, demonstrating beneficial effects. Similarly, *Bacteroidales* and *Enterobacteriales* have been described to have both beneficial and detrimental effects [142]. Observational cohort studies involving humans have also identified differences in gut microbiota composition between individuals with and without FA, suggesting that distinct microbiota profiles may have varying effects on food allergen tolerance [143,144,145]. Moreover, direct profiling of the gut microbiome in these studies has shown that dysbiosis, characterized by microbial imbalance, occurs before the onset of FA in affected individuals.

Microbiota–host interactions play a key role in regulating the immune system. The interplay between the gut microbiome and the development of FA is thought to occur through the immunomodulatory effects of microbes on food allergen tolerance [146]. The gut microbiome promotes tolerance by inducing the retinoic orphan receptor gamma T (ROR-γt)^+^ regulatory T-cells, which can be facilitated by the microbial production of short-chain fatty acids, including butyrate [147]. Furthermore, the intestinal microbiota may prevent allergic sensitization to food antigens through stimulating the production of IL-22 by immune cells in the gut, resulting in improved intestinal epithelial integrity and reduced interaction of the immune system with the allergen [148].

Early-life insults to the gut microbiota impair the differentiation of ROR-γt+ regulatory T (T_reg_) cells and disrupt the development of oral tolerance to food antigens. Notably, this dysregulated immune response is characterized by decreased IgA binding and increased IgE binding to the gut microbiota, indicating impaired anti-microbial antibody responses. Moreover, T follicular helper cell (Tfh) responses to food antigens develop, thereby facilitating the production of high-affinity IgE antibodies that can trigger anaphylactic reactions.

Understanding the intricate interactions between leaky gut, gut microbiome, and FA has significant implications for prevention and treatment strategies. Therapeutic interventions that restore gut barrier integrity and rebalance the gut microbiota composition and functionality hold promise in ameliorating FA symptoms. Approaches such as the use of specific probiotic strains, prebiotics, synbiotics, postbiotics, and dietary modifications have shown potential in preclinical and clinical studies by promoting a healthy gut microbiota, enhancing gut barrier function, and modulating immune responses, thereby reducing the risk of FA and improving the management of existing allergies.

## 2. Therapeutic Interventions to Restore Intestinal Barrier Integrity and Microbiome in Food Allergies

### 2.1. Diet, Prebiotics, and Short-Chain Fatty Acids

Prebiotics usually comprise oligosaccharides and short-chain polysaccharides, serving as nutritional substrates for beneficial gut microbial bacteria, promoting their expansion [149]. Fructans, specifically inulin and fructooligosaccharides (FOS), and galactans such as galactooligosaccharides (GOS) are the most studied prebiotics, since they are found in foods such as some vegetables (garlic, chicory, onion, artichoke, and asparagus), banana, and cereals such as rye and corn.

Human milk also comprises prebiotics named human milk oligosaccharides (HMOs). One- to two-week interventions with supplementation of inulin [150,151,152,153], FOS [154], or GOS [154,155,156,157] were significantly associated with an increase in abundance of *Bifidobacterium*, as well as *Lactobacillus*, *Akkermansia*, or *Roseburia*. *Bifidobacterium* spp. are the main commensals for HMO degradation [158]. Maternal prebiotic supplementation during pregnancy to prevent allergy in children seems to be the best window of opportunity to modulate gut microbiota composition and functions [159].

Gut bacteria ferment prebiotics and induce the production and release of short-chain fatty acids (SCFAs) in the lumen [50,160]. SCFAs can specifically interact with intestinal epithelial cells and innate/adaptive immune cells to influence cellular differentiation, proliferation, and apoptosis [161]. SCFAs induce signaling pathways through the stimulation of the production of protein kinases [162], mammalian target of rapamycin (mTOR) [163], and nuclear factor kappa-light-chain-enhancer of activated B cells (NFκB) [164], while SCFAs also modulate the function of histone acetyltransferase (HATs) and histone deacetylases (HDACs) involved in the regulation of gene expression [165]. In preclinical studies, butyrate significantly reduced food allergic response, with induction of tolerogenic cytokines, inhibition of Th2 cytokine production, and modulation of oxidative stress [166]. A receptor for butyrate and niacin (GPR109A) may have a crucial role in maintaining epithelial function and may represent a negative regulator of immune responses [167]. Recently, Wang et al. showed that treatment with butyrate-releasing micelles increased the abundance of butyrate-producing taxa in *Clostridium* cluster XIVa and protected mice from an anaphylactic reaction to peanut challenge [168].

Interestingly, oral immunotherapy and FOS treatment in mice induce potential microbial alterations associated with increased butyrate levels in cecum content, representing promising strategies to improve oral immunotherapy efficacy in human studies to treat FA [169]. In humans, recent studies suggest that butyrate may directly affect mast cells by epigenetically regulating the FcεRI-mediated signaling molecules [170]. Thus, butyrate could have therapeutic benefits in human FA by inhibiting IgE-mediated mast cell degranulation and allergen-induced histamine release [171]. Moreover, in human enterocytes, butyrate has been shown to stimulate mucin production, tight junctions, and human beta-defensin-3 expression. In peripheral blood mononuclear cells from children with FA, butyrate enhanced IL-10, IFN-γ, and Forkhead box P3 (FOXP3) expression through epigenetic mechanisms [166]. Moreover, it promoted dendritic cells, regulatory T cells (T_reg_s), and the precursors of M2 macrophages [166].

All these recent findings highlight the critical role of prebiotics and SCFAs in promoting the integrity of the epithelial barrier, oral tolerance, and protection against FA. These observations could, at least in part, be explained by the inhibitory effects of SCFAs on HDACs in various immune cells, such as T_reg_s, B cells, and mast cells, as well as via stimulation of various SCFA receptors. However, recent studies [172,173] showed that prebiotic supplementation (especially inulin) at a high dose exacerbated the allergic inflammation in the context of excess accumulation of SCFAs and microbiota dysbiosis, emphasizing the importance of personalized use of prebiotics to safely alleviate FA. Indeed, interspecific interactions between gut microbial bacteria remain unclear and should be considered. The possible harmful effects of excessive proliferation of high-SCFA-producing bacteria should not be neglected.

### 2.2. Probiotics, Next-Generation Probiotics, and Postbiotics

Probiotics are live microorganisms that may benefit the host’s health when administered in adequate amounts [174]. Even if this term also includes yeasts, such as *Saccharomyces boulardii* [175], the most important probiotics studied in FA are bacteria falling into two genera, namely *Lactobacillus* and *Bifidobacterium*. The exact mechanism of action is still little understood; however, probiotic supplementation may prevent or reduce FA symptoms in several ways. First, they inhibit the blooming of pathogenic microorganisms competing with them for nutrients and producing antimicrobial metabolites; they adhere to intestinal epithelial cells through surface glycoprotein or polysaccharides, hampering the binding of pathogenic bacteria to epithelial cells. In addition, lactic acid bacteria (LAB) and *Bifidobacteria* contribute to maintaining a low oxygen tension and an acid gut environment, as an ideal habitat for beneficial species [176].

For this purpose, supplementation with *L. rhamnosus* GG (LGG) during a dietary intervention with extensively hydrolyzed casein formula (EHCF) in infants affected by cow milk allergy (CMA) was associated with the expansion of *Blautia*, *Roseburia,* and other butyrate-producing taxa and with an increase in fecal butyrate levels [177]. Probiotics enhance the gut barrier, producing both SCFAs as their main metabolites (see above) and increasing the expression of mucins (MUC2, MUC3, and MUC5AC) by intestinal cells [178]; hence, they enforce the expression of tight junction proteins such as occludins and zonulin protein (Zonula Occludens-1, ZO-1) [179]. In this way, they reduce epithelial barrier derangement, which is the main cause of bacterial and peptide translocation into the inner barrier, leading to microinflammation and immunization (the so-called “epithelial barrier hypothesis” of the onset of allergic diseases) [176].

As the basis of most food allergic reactions, there is an immune imbalance during the gut-associated lymphoid tissue (GALT) development toward a Th-2 phenotype. Whereas, in utero, the Th2 pattern is predominant to prevent maternal rejection, in the early phases of post-natal life, microbial stimulation, including vaginal bacteria such as *Lactobacilli*, reprograms the immune system towards a Th1 profile [16]. As demonstrated in a murine model of FA to ovalbumin (OVA), cesarean section (CS) yields a lack of *Lactobacillus* and *Bifidobacterium* strains in the feces and worsens the severity of allergic symptoms. Probiotic supplementation, based on a mixture of *L. acidophilus*, *B. longum* subsp. *Infantis*, *E.faecalis*, and *B.cereus* from birth to day 21 could rescue CS-related dysbiosis, improve allergic symptoms, reduce IgE and IgG production, and increase the expression of tight junction protein such as ZO-1, occluding, and claudin-1 [180]. These results are in line with those obtained in other animal studies, showing that new-born mice in antibiotic therapy and germ-free mice maintain a Th2 immune phenotype and an increased risk of allergic reactions to oral antigens. This risk is often corrected by probiotic supplementation in early life, which restores the Th1/Th2 balance, favoring a Th1 pattern [181,182,183].

Such results have been confirmed in the last two decades by several human studies, both retrospective and prospective (non-randomized and randomized trials), mostly in the management of CMA [176]. Most of the research has focused on *L. rhamnosus* GG, *L. casei*, *L. plantarum*, *B.lactis*, *B. bifidum*, *B. breve,* and included infants affected by CMA and orally supplemented with probiotics over a period ranging from at least 1 week [184] to 36 months [185]. Depending on the type of study, children were then followed for a period ranging from 72 h [184] to 5 years [186]. These studies had different endpoints, from reductions in daily vomiting and diarrhea in the early phases of supplementation [184] to the achievement of clinical food tolerance at 12 months [187]. Some of them used a unique strain such as *L. rhamnosus* GG [188] or a mixture of *L. rhamnosus* GG and *B. lactis* [187]. Others also included prebiotics like inulin and oligofructose [189]. Almost all these studies confirmed a significant impact of probiotic supplementation in ameliorating clinical features and reducing allergic reactions in infants affected by CMA, thus showing possible benefits. However, current evidence does still not support a clear recommendation on the use of probiotics in the prevention or treatment of FA by scientific societies, lacking clear information on the specific strain, dosage, and adequate duration of therapy [16].

A possible novel field of interest is oral food immunotherapy (OIT), based on the idea that progressive and continued oral or intestinal exposure to dietary antigens may lead to food tolerance [190]. In this concept, probiotics may figure as immune response modifiers (IMR), a class of compounds of microbial origin (bacteria themselves) that modulate immune response, together with antigen-presenting cells [16]. In a mice model of egg allergy, animals fed with *L. casei* variety *rhamnosus* (Lcr35) had a reduced anaphylactic response during OIT compared to controls (no OIT or only OIT) [191]. A randomized control trial in children affected by peanut allergy showed sustained unresponsiveness in patients treated with probiotics (*L. rhamnosus* CGMCC 1.3724) and peanut OIT (82.1%) compared to the placebo group (only peanut OIT) [192]. However, a recent multicenter randomized trial did not confirm these findings, showing similar rates of sustained unresponsiveness in both OIT and OIT + probiotic (*L. rhamnosus* ATCC 53103) supplementation [193].

Next-generation probiotics (NGP) are an emerging field of research in FA treatment. Differently from the traditional types, these microorganisms have been determined using next-generation sequencing techniques and bioinformatics methods. NGP may also include postbiotics, intended as “non-viable bacterial products or metabolic products from microorganisms that have biologic activity in the host” [194]. These microorganisms may also increase the shelf life of probiotic products and are generally safe when administered to immunocompromised individuals [195].

*Akkermansia muciniphila* is the prototype of NGP and the first approved by the European Food Safety Authority (EFSA) [196]. Its action is also exerted in non-vital cells through its surface protein (Amuc_1100) and has been extensively documented in diabetes mellitus type-2 and obesity [197]. Evidence shows that *A. muciniphila* is able to produce SCFAs and stimulate IL-10 synthesis and T_reg_ cell proliferation, hence reinforcing the intestinal barrier [198]. In a murine model of OVA, the supplementation of viable *A. muciniphila* reduced serum levels of anti-OVA IgE and body weight loss (marker of illness) in allergic mice and reduced intestinal inflammation and leukocyte infiltration compared to controls. Furthermore, the inactivated bacteria significantly reduced anti-OVA IgE levels and eosinophil recruitment in allergic mice. This also confirmed the role of postbiotics in this setting [199]. Similar results were obtained in the same model through the supplementation of *Bifidobacterium longum* subsp. *longum* 51A (BL51A) only as viable cells, with the results depending on the dose and viability of the bacteria. In this case, an increase in the anti-inflammatory IL-10 was obtained. However, treatment with inactivated bacteria had no beneficial effects [200].

All these results confirm the potential role of probiotics and postbiotics in the management of FA (Table 1). However, further studies on the types of strain, dosage, and duration of treatment are still warranted to consider them as part of therapy.

### 2.3. Fecal Microbiota Transplantation

Fecal microbiota transplantation (FMT) is defined as the infusion of stool from a healthy donor to the gastrointestinal (GI) tract of a recipient patient, with the aim to modifying the gut microbiota and restoring the condition of dysbiosis [201]. FMT can be performed via colonoscopy, oral capsules, or nasogastric tube [202]. Nowadays, FMT has proven its efficacy as a potential alternative to antibiotics in *Clostridioides difficile* infections, and it has also been integrated into international guidelines [203]. An increased body of evidence suggests the use of FMT in other disorders, including GI disorders (i.e., IBD and irritable bowel syndrome) [204], liver disease [205], metabolic disorders [206], and malignancies [207].

FMT has been proposed as a treatment in FA, due to its ability to modulate gut microbiota. This proposal stems from studies that have shed light on the role of different gut microbiota compositions in these patients [208].

Preliminary preclinical evidence suggests a potential role for FMT in FA. Two papers reported a decrease in the core body temperature in mice receiving FMT from FA-affected patients, while only a temporary drop was observed for mice receiving FMT from healthy donors, indicating a protective role against anaphylaxis after oral food challenge [140,141]. Moreover, higher fecal scores and increased GI symptoms (scratching, diarrhea, etc.) were observed in mice who were infused feces from FA when compared with the healthy control mice, when they underwent oral allergic sensitization [209]. Additionally, lower alpha and beta diversity; decreases in *Bacteroidaceae* and *Lachnospiraceae* families; and increases in *Clostridiaceae*, *Enterobacteriaceae*, and *Bifidobacteriaceae* were observed among mice receiving FMT from patients with FA and from healthy donors [210].

The emerging role of FMT in FA and the findings obtained in mice models have opened the road to new studies in humans. A Phase I open-label trial to evaluate the safety and tolerability of oral encapsulated FMT administered over 2 days for the treatment of peanut allergy in 10 adult subjects (18–40 years) is now closed, but the data have not been published yet [211]. Moreover, a phase II randomized double-blind placebo-controlled trial to evaluate the safety and tolerability of oral encapsulated FMT in 24 peanut allergic patients (12–17 years) is ongoing [212].

## 3. Conclusions

Increasing evidence, as reviewed in this manuscript, clearly supports a role for gut microbiota imbalance and disruption of the intestinal barrier in the pathogenesis of FA.

These pathogenic pathways are elicited by environmental factors, including industrialization or unbalanced diet (i.e., with the consumption of highly processed food), that may trigger Th2 immune responses and increase the susceptibility to allergic sensitization.

Preliminary evidence from preclinical and clinical experiences supports the potential use of various modulators of gut microbiome, including diet, prebiotics, probiotics, postbiotics, and FMT, as potential therapeutic tools for FA, although further well-designed studies are needed to confirm these insights.

## Figures and Tables

**Figure 1 nutrients-16-00092-f001:**
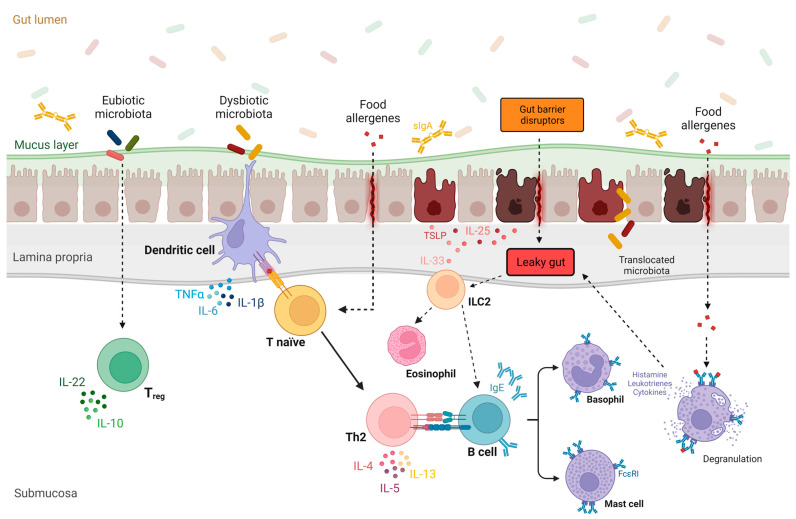
The role of gut microbiota and leaky gut in the pathogenesis of food allergy. Under physiological conditions, a eubiotic gut microbiota promotes the differentiation of T lymphocytes into T regulatory cells, leading to immune tolerance of the gut [96]. Conversely, in the context of a leaky gut barrier and intestinal dysbiosis, epithelial-derived cytokines such as thymic stromal lymphopoietin (TSLP), interleukin 33 (IL-33), and IL-25 are released in response to various gut barrier disruptors [97,98]. These alarmins promote a pro-allergic microenvironment by activating T helper 2 (Th2) and type 2 innate lymphoid cells (ILC2), resulting in the release of proinflammatory cytokines (e.g., IL-4, IL-5, and IL-13) [95]. Furthermore, dysbiotic gut microbiota induce the differentiation of Th2 cells, promoting the IgE class-switching process in B cells [100]. After sensitization to a specific food allergen, which is also favored by leaky gut, allergen-specific IgE antibodies become immobilized on the surface of basophils and mast cells. Upon subsequent exposure to the allergen, these cells release histamine and other proinflammatory mediators (e.g., leukotrienes and type 2 cytokines) [101,102], worsening gut permeability and amplifying type 2 inflammation [103]. Abbreviations: FcεRI, high-affinity IgE receptor; IL, interleukin; IgE, immunoglobulin E; ILC2, innate lymphoid cell2; Th0, naive T cell; Th2, helper T cell 2; T_reg_, T regulatory cell; TLSP, thymic stromal lymphopoietin; sIgA, secretory IgA.

**Table 1 nutrients-16-00092-t001:** Potential roles of prebiotics/probiotics in alleviating or preventing food allergies.

	Prevention of Gut Barrier Integrity	Regulation of Immune Response	Anti-Inflammatory Effect
**Prebiotics** HMOs	Promotion of beneficial bacteria growth (such as *Bifidobacterium, Lactobacillus*, *Akkermansia*, or *Roseburia)*	Promotion of Th1/Th2 balance Induction of tolerogenic cytokines	↓ Inflammation ↑ Butyrate-producing bacteria
**Prebiotics** FOS/GOS/Inulin	Regulation of mucin production Stimulation of beneficial bacteria growth ↓ Allergen and pathogen translocation	↑ T_reg_ number and function ↑ TNF-α	↑ SCFAs production (especially butyrate) ↓ Inflammation
**Probiotics ***	↓ Bacterial translocation ↑ Tight junction protein expression ↑ Epithelial integrity ↑ Expression of mucins by intestinal cells	Restoration of Th1/Th2 balance Downregulation of inflammatory cytokines (such as IL-6, IL-8 and TNF-α) ↑ T_reg_ numbers and function ↑ IL-10 e TGF-β ↓ allergen-specific IgE ↑ allergen-specific IgG	↑ Anti-inflammatory mediators ↓ Inflammatory cells ↑ SCFAs production (especially butyrate)

* lack of clear information on the specific strain, dosage, and adequate duration of probiotic therapy. Abbreviations: ↑, increase; ↓, decrease; FOS, fructo-oligosaccharides; GOS, galacto-oligosaccharides; HMO, human milk oligosaccharides; IFN, interferone; IL, interleukin; SCFAs, short-chain fatty acids; TGF, transforming growth factor; TNF, tumor necrosis factor.

## Data Availability

Not applicable.

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
