# Peer review of "The Role of Gut Microbiota and Leaky Gut in the Pathogenesis of Food Allergy"

_nutrients, 2023, doi:10.3390/nu16010092_

Round 1
Reviewer 1 Report
Comments and Suggestions for Authors
The authors provided an overview of the relationship between gut microbiota, intestinal permeability, and food allergies, as well as the potential of probiotics, prebiotics, and fecal microbiota transplantation (FMT) in prevention and treatment. It is acceptable with minor revisions.
1.The author's abstract mentions that environmental factors, such as industrialization and consumption of highly processed food, can alter the gut microbiota and intestinal barrier, increasing susceptibility to allergic sensitization. However, there is limited content related to this issue in the article.
2. The author states that changes in the composition and function of the gut microbiota are associated with food allergies, and it is possible that food allergies may cause such changes. However, does alteration of the gut microbiota and intestinal permeability caused by other factors also increase the risk of food allergies?
3. In Section 1.4, the author describes many factors can cause changes in intestinal permeability and the dysregulation of inflammatory factors/related proteins during this process. Does food allergy lead to the same changes? Do the mechanisms involved in these changes remain the same? This requires further clarification.
4. Based on Section 2, the author should create a table that includes types of prebiotics/probiotics and their roles in alleviating or preventing food allergies.
5. The author mentions the use of next-generation probiotics, which is interesting and meaningful. Besides the ones mentioned by the author, are there any other types of NGPs? If so, the author should provide additional information.
Comments on the Quality of English Language
Minor editing of English language required
Reviewer 2 Report
Comments and Suggestions for Authors
The literature review by Poto and colleagues on the role of Gut microbiota and "leaky gut" in the pathogenesis of food allergy, is a thorough and nicely written review.
-I would just omit the term "imbalance" from the title, as this is difficulty to define. "The role of gut microbiota.." should be enough, as the microbiota of individuals with FA has been reported to be different from that of healthy or non-allergic individuals.
-Please add relevant recent literature: Parrish A. et al., 2023 and Courtney Hoskinson et al., 2023.
-I would add the work by Dr. Caminero in celiac disease, that shows that the microbiome can alter antigen immunogenicity, and is thus applicable to other conditions, such as FA.
Reviewer 3 Report
Comments and Suggestions for Authors
It was a comprehensive review worth reading. It is well discussed in terms of content, but it would be more cohesive and more comprehensible to the reader if there were figures and tables in the references presented here and elsewhere. Please consider whether figures and tables could be added to the content again.
